# Efficacy of Ultrasound-Guided Percutaneous Lavage and Biocompatible Electrical Neurostimulation, in Calcific Rotator Cuff Tendinopathy and Shoulder Pain, A Prospective Pilot Study

**DOI:** 10.3390/ijerph19105837

**Published:** 2022-05-11

**Authors:** Raffaello Pellegrino, Angelo Di Iorio, Cristina Maria Del Prete, Giovanni Barassi, Teresa Paolucci, Lucrezia Tognolo, Pietro Fiore, Andrea Santamato

**Affiliations:** 1Antalgic Mini-Invasive and Rehab-Outpatients Unit, Department of Innovative Technologies in Medicine & Dentistry, University “G. d’Annunzio”, Viale Abruzzo 322, 66100 Chieti, Italy; raffaello.pellegrino@ucm.edu.mt (R.P.); giovanni.barassi@unicatt.it (G.B.); 2Department of Scientific Research, Campus Ludes, Off-Campus Semmelweis University, 6912 Lugano, Switzerland; 3Department of Physical Medicine and Rehabilitation, Azienda Sanitaria Locale, 73100 Lecce, Italy; cridelprete@tin.it; 4Center for Physiotherapy, Rehabilitation and Re-Education-CeFiRR-Gemelli Molise, 86100 Campobasso, Italy; 5Physical Medicine and Rehabilitation, Department of Oral Medical Science and Biotechnology, University “G. d’Annunzio”, 66100 Chieti, Italy; teresa.paolucci@unich.it; 6Rehabilitation Unit, Department of Neuroscience, University of Padova, 35122 Padua, Italy; lucrezia.tognolo@unipd.it; 7Neurological Rehabilitation and Spinal Unit, 70124 Bari, Italy; pietro.fiore@unifg.it; 8Spasticity and Movement Disorders “ReSTaRt” Unit, Physical Medicine and Rehabilitation Section, OORR Hospital, University of Foggia, 71122 Foggia, Italy; andrea.santamato@unifg.it

**Keywords:** barbotage, calcific tendinopathy, biocompatible electrical neurostimulation, shoulder pain

## Abstract

Calcific tendinopathy of the shoulder (CTS) is the most common cause of shoulder pain. Conservative treatment is considered as the first therapeutic choice for CTS. The main objective of this study was to assess the effect of US-guided needling (UGN) compared to UGN plus Biocompatible Electrical Neurostimulation (BEN) in the treatment of the CTS. Pilot, prospective, non-interventional, monocentric, and observational study of patients treated for calcific rotator cuff tendinopathy and shoulder pain. Patients’ selection, enrollment and interventions were conducted at the Chiparo Physical Medicine and Rehabilitation outpatient facility. Forty adult patients (aged 40–60 years) with a diagnosis of CTS in the acute and colliquative phase were recruited and enrolled into the study. Participants were assessed for self-perceived pain through the Numerical Rating Scale (NRS), and for functional limitation through the Shoulder Pain and Disability Index score (SPADI) at baseline (T0), after 15 days (T1), and after 40 days (T2). As a possible confounding factor between the two treatments’ response, the dimension of the tendon calcification was also assessed by US-examination. Through the study, both groups improved their perceived functional performance of the arm (*p*-value < 0.001). AT T1, the SPADI score decreased by half in both groups, and the improvement remained stable at T2. A multiplicative effect (Time × Treatment) was demonstrated (*p*-value < 0.001). An improvement in the NRS score was measured at T1, and it remained stable at T2, a multiplicative effect was also reported (*p*-value < 0.001). The main results of this pilot study provide evidence that UGN plus BEN increases functional performance and reduces shoulder pain in individuals with CTS. Moreover, the tendon calcification dimension at the baseline and the percentage of drainage of the lesion were associated with a functional performance recovery and pain reduction detected after intervention.

## 1. Introduction

Calcific tendinopathy of the shoulder (CTS) is the most common cause of shoulder pain. It is characterized by the presence of calcium-hydroxyapatite crystal located within the tendons of the rotator cuff [1]. CTS mainly affects middle-aged women and it can be asymptomatic in the 35–50% of cases [2]. There are two forms of CTS: the enthesopathic, and the intratendinous. The enthesopathic form presents calcifications typically localized at the tendon insertion, often associated with erosions of the tendons [3]. The most frequently affected tendon (80%) is the supraspinatus, followed by the infraspinatus (15%) and the subscapularis (5%) [3]. The etiology and pathophysiology of CTS are not yet fully understood, although the multifactorial pathogenetic theory, based on a cell-mediated pathogenetic mechanism and associated with metabolic factors (thyroid hormones, diabetes mellitus, hyperlipidemia), a genetic predisposition to the formation of calcium deposits, seems to be gradually gaining acceptance [1]. CTS is a self-limiting disease, characterized by three main stages: the pre-calcium, the calcium (divided in turn into three sub-phases), and post-calcium stage [4]. The first stage, often asymptomatic, is characterized by a fibro-cartilage metaplasia of the tenocytes, related to a particular histocompatibility antigenic pathway, Human Leukocyte Antigens 1 (HLA1) [5], with a consequent increase in the production of proteoglycans and the formation of fibrocartilage tissue [6]. During the calcium stage, the calcium-hydroxyapatite crystals are deposited in vesicles scattered between the chondrocytes and the fibro-cartilage tissue, making larger deposit areas free of vascularization [4]. Macroscopically, the calcification at this stage is characterized by a chalky appearance. The third stage, the post-calcic (or resorptive), is characterized by an increase in vascular neoformation with thin walls that allows macrophages and multinucleated giant cells to surround the calcifications [4]. Macroscopically, in this phase, the calcific tissue is whitish and dense, with a consistency similar to a cream or a toothpaste [7].

Clinically, the CTS presents with a different grade of severity and pattern of symptoms depending on the stage, size, and location of the calcifications. The clinical picture is characterized by referred pain localized in the anterior region of the shoulder or radiated to the posterior area of the scapula or upwards to the neck. The presence of pain is characteristic in the reabsorption/colliquation phase of the calcification [1,4]. The histomorphology modifications of the CTS induce a volumetric growth in the size, with an increase in the intratendinous pressure. A background local inflammation state is also reported in the literature, with a considerable mainly overnight pain, accompanied by functional limitation, and by the symptoms typical of the sub-acromial conflict syndrome [8]. Plain shoulder radiographs, ultrasonography, and MRI are essential in the diagnostic processes, although the ultrasound (US) scan study clearly has some diagnostic advantages [9]. Moreover, US study appears to be an exhaustive tool in identifying the localization, size and stage of CTS, as well as providing information on the qualitative aspects of the tendons and bursas. Using the US, we can distinguish the “dome phase”, with the typical posterior acoustic shadow, also called the state phase (Appendix A), from the colliquative phase, also called the hyperalgic phase, characterized by a thin peripheral hyperechoic rim and a hypoechoic or anechoic core [10], blurred edges into the subdeltoid bursa (Appendix A) [11]. There is a general agreement in considering the conservative treatment as the first therapeutic choice for CTS. Non-steroidal anti-inflammatory drugs (NSAIDs) and analgesics are indicated to control pain during the hyperalgic phase [12] when the therapeutic exercise and the joint mobilization might, on the contrary, increase intra-tendon pressure, exacerbating the pain [13]. In the acute phase, if the patient has bursitis or impingement symptoms, a treatment with anesthetics and corticosteroids can also be used by delivering them into the sub-acromial space [14]. Additionally, classified as a conservative treatment, US-guided needling (UGN) is indicated in the therapeutic management of the colliquative and hyperalgic phase of the calcific tendinopathy [9]. Another possible conservative intervention, in the hyperalgic phase, is the new generation of Biocompatible Electrical Neurostimulation (BEN). This approach aims at promoting better pain control, stimulating regeneration processes and modulating the neo-angiogenesis [15]. BEN is based on a monophasic, asymmetrical, biocompatible pulse, with an active phase using high negative voltage, a very short duration (5–100 μs) and a variable frequency (3–875 Hz). The high negative voltage allows the depolarization of the cell membrane and opens the ion channels. The depolarization re-activates the recovery of the state of natural homeostatic equilibrium through a “reset” mechanism. BEN treatment is composed of basic pulse sequences with different frequency, amplitude and duration (multi-parametric modulation). The clinical effects show a significant reduction in pain, tissue regeneration and functional recovery of the treated area [16]. UGN and BEN could have a synergic effect when used in combination, thanks to mechanisms suggesting their therapeutic effectiveness. Moreover, there is a lack of studies with a high level of evidence comparing combined treatment methods and the preferred treatment for CTS remains a subject of debate.

Therefore, the main objective of this study is to assess the effect of UGN (standard of care), compared to UGN plus BEN (treatment group), in the treatment of calcific rotator cuff tendinopathy and shoulder pain.

## 2. Materials and Methods

This was a pilot, prospective, non-interventional, monocentric, observational study of patients treated for calcific rotator cuff tendinopathy and shoulder pain with UGN plus BEN (combined-treatment group, CoTreatG), compared with UGN-alone (UGN-A).

The study design was conceptualized at the University “G. d’Annunzio”. The patients’ selection, enrollment and the interventions were conducted at the Chiparo Physical Medicine and Rehabilitation outpatient facility, Lecce, Italy.

Between March 2019 and November 2020, adult patients aged 40 to 60 years, with a diagnosis of CTS, in fluid calcifications, hypo or anechoic, without acoustic shadowing, were recruited and enrolled into the study (stage III according to Sconfienza classification) [10]. Among those subjects who were offered to be part of the study, no one declined to participate, nor did withdrawal due to serious, moderate, or light adverse events occur. The enrolled individuals met all inclusion criteria and none of the exclusion criteria.

Enrollment was offered to adult subjects over 18 years of age, affected by calcific tendinopathy of the shoulder rotator cuff in the hyperalgic and colliquative phase of calcification, with ultrasound features of without acoustic shadowing and blurred margins of the calcification. Participants were willing and able to give informed consent for the participation in the study.

Patients were excluded if they had: (1) a poor compliance in attending the follow-up visits; (2) a diagnosis of rupture of any shoulder rotator cuff tendon; (3) a diagnosis of diabetes mellitus; (4) hypertension not well pharmacologically controlled; (5) drug hyper-sensibility; (6) a diagnosis of epilepsy; (7) a diagnosis of coagulopathies; (8) a current diagnosis of neoplasms; (9) an established pregnancy; (10) shoulder skin infections around the intervention area; (11) high temperature detected on the day of the procedure; (12) an implanted pacemaker; (13) a previous or current history of panic attacks. Moreover, subjects with the following comorbidities were also excluded: fibromyalgia, seronegative entesoarthritis, rheumatoid arthritis, or if they were on oral anticoagulant therapy or chronic use of opioid drugs.

The study was developed following the Good Clinical Practice (GCP) guidelines. It was conducted within the ethical principles outlined in the Declaration of Helsinki, and with the procedures defined by the ISO 9001-2015 standards for “Research and experimentation”. Written informed consent was obtained at baseline from all participants. The data collection and the procedures applied are parts of the standard clinical routine, therefore, the normal ethics committee clearance was not required.

The sample size of 40 patients was calculated with a Control group Event Rate (CER) of two points on the NRS score outcome and a CER of three points on the Visual Analogue Scale outcome. In addition, 0.05 α-error (Type-I), 0.20 β-error (Type-II).

### 2.1. Outcome Measures

All subjects underwent a clinical and anamnestic evaluation; an US examination of the shoulder was performed, as well as a disability evaluation assessed by the Shoulder Pain and Disability Index (SPADI) questionnaire [17,18]. The SPADI questionnaire, a patient self-completed district assessment tool commonly used to assess shoulder musculoskeletal disorders, was also applied. The questionnaire consists of 13 items divided into two modules designed to assess shoulder pain (5 items) and disability (8 items). The answers are indicated on a visual analogue scale where 0 is no pain/no difficulty and 10 is worst imaginable pain/so difficult it requires help. The items are summed and converted to a total score out of 100.

Pain was assessed by the use of the Numerical Rating Scale (NRS score) [19] that varies between 0 and 10 with 0 indicating no pain and 10 indicating the worst possible pain. As a possible confounding factor between the two treatments’ response, the dimension of the tendon calcification was also assessed by US examination. Outcomes’ measures were assessed at baseline (T0), at day 15 (T1) during the intervention, and at the end of the intervention at day 40 (T2).

### 2.2. Percutaneous Ultrasound-Guided Lavage

Both groups underwent the mini-invasive procedure of percutaneous ultrasound-guided lavage of the intratendinous calcifications of the shoulder. The subjects were positioned as semi-seated on the seat with the arm flexed at 90° of the elbow and internal rotation of the shoulder. Following disinfection of the supradeltoid skin surface, a subacromial intrabursal ultrasound anesthesia was performed using 3 cc of lidocaine (2%) with a 22 G 40 mm needle inclined at 60° from the skin surface, with a visual check of the progression of the needle until the sub-acromial border (Appendix A). Subsequently, using a 5 mm 25 G needle, a further 3 cc of 2% lidocaine was administered in small wheals in the skin area identified as the spot for the injection of the washing needles. Seven minutes following the anesthesia injection, three needles (16/18/G) were inserted, with a 60-degree angle from the skin surface, with the position of the needles checked by US until the intratendinous calcification was achieved (Appendix A). The three 40 mm needles were placed into the calcification while they were communicating with each other. The next phase of the therapeutic procedure involved the insertion of a small connector in alternative succession to each needle (Appendix A). This connector was subsequently attached with a 30 cc syringe containing saline solution. The procedure continued with a succession of lavage and aspirations with room-temperature saline and lidocaine 2% in the context of the calcification, in order to remove the calcium residues from the tendon (Appendix A). At the end of the lavage phase, an US guided intrabursal injection of 40 mg of triamcinolone acetate was also performed. 

### 2.3. Biocompatible Electrical Neurostimulation

Within the 48 h following the percutaneous treatment, patients started a daily BEN treatment, consisting of ten sessions of 20 min each, with the LIFE-Stim™ (EME srl, Pesaro, Italy) variable frequency electrical transcutaneous analgesic modulation device. The treatment was administered regularly, and titrated up to remain perceived as ‘strong but comfortable’ during use. Patients were seated with the upper limb positioned in a neutral position of internal rotation and elbow flexion. The two pairs of electrodes of channel 1 were positioned in the anterolateral region of the deltoid with the red electrodes proximally and the black electrodes positioned more distally. Conversely, the first pair of electrodes of channel 2 was placed in the posterior lateral area of the deltoid, with the red electrode placed at the posterior edge of the acromion process and the black electrode placed 5 cm posteriorly. The second pair of electrodes of channel 2 was placed parallel to the first pair, but 8–10 cm lower (Appendix A).

### 2.4. Statistical Analysis

Data were reported as mean ± standard error (S.E.) for continuous variables and as absolute number and percentage for dichotomous variables; differences between groups were assessed with analysis of variance and chi-square test, respectively. To take into account the role of potential confounders, multivariate models were also conducted. To assess the variation in SPADI score and NRS score during the study, Linear Mixed Models (LMM) were applied [20]. Intercept and Time had a random component. The advantage of this approach is that it increases the precision of the estimate by using all available information concerning performance and, at the same time, allows for handling missing data, and had a more powerful modelling of the analysis. From estimates computed in the LMM, we calculated: ρ = intraclass coefficient correlation and $R_{e}^{2}$ = pseudo-R2 statistic that assess the proportion of within-person variation explained by time [20]. Data were analyzed with SAS software (SAS/STAT^®^ software, rel. 9.4, SAS Institute Inc., SAS Campus Drive, Cary, NC, USA), and *p*-value for differences was considered statistically significant for a value less or equal to 0.05.

## 3. Results

Forty patients (24 women, 16 men) were enrolled, 20 for each group. The mean age for the study population was 40.3 ± 6.0 (49.1 ± 5.5; 49.4 ± 6.3; for experimental and control, respectively, *p*-value = 0.85). No statistically significant differences could be found for sex between the two groups; females were 11 (55.0%) and 13 (65.0%) for treatment and control, respectively (*p*-value = 0.56). The treatment group reported at baseline, a higher functional limitation, recorded with the SPADI score (*p*-value = 0.03), and a higher level of NRS score self-reported pain (*p* = 0.04), compared to the control group (Figure 1 and Figure 2), independently from age and sex.

During the study, both groups improved their precepted functional performance of the arm (time for trend *p*-value < 0.001); as matter of fact, at T1 follow-up the SPADI score was practically reduced by half in both groups, and the improvement remained stable at T2. A multiplicative effect was demonstrated by the statistically significant interaction for time and treatment (Time × Treatment *p*-value < 0.001). Moreover, 4.5% in the SPADI score is attributable to differences between groups (ρ = 4.5%), whereas 44% of the within-person variation in SPADI score is associated with linear time ($R_{e}^{2}$ = 44.1).

A similar figure was demonstrated for NRS self-reported pain (Figure 1). Similarly, NRS score decreased at T1, and remained stable at T2, and again a multiplicative effect was reported (Figure 2). Moreover, seven percent in the NRS score is attributable to differences between groups (ρ = 7.1%), whereas 41% of the within-person variation in NRS score is associated with linear time ($R_{e}^{2}$ = 40.7).

At the baseline, the calcification sizes were very similar (CoTreatG: 1.43 ± 0.23 cm; BEN 1.38 ± 0.25 cm; *p*-value = 0.50); moreover, from T0 to T2 assessment, a statistically significant reduction in the dimensions for all the patients enrolled in the study (−0.77 ± 0.03; *p*-value < 0.001) was found (Appendix A). No multiplicative effect for the interaction between T0, T1 and T2 and treatment group were detected (0.03 ± 0.07; *p*-value for the interaction = 0.70). Lastly, 5 patients (12.5%) out of the 40 enrolled showed a percentage of lesion reduction ranging from 35% to 45%, while 13 (32.5%) patients showed a percentage of lesion reduction above the 60%.

The tendon calcification dimension was associated with both perceived functional performance of the arm by mean of SPADI score (1.99 ± 0.11; *p*-value < 0.001) and the self-reported pain NRS score (4.68 ± 0.30; *p*-value < 0.001) independently from group treatment and evaluation time; for the same variables there was no multiplicative effect for the interactions.

## 4. Discussion

The main results of this pilot intervention, monocentric, open-label clinical trial controlled with standard of care provide evidence that Biocompatible Electrical Neurostimulation increases functional performance and reduces shoulder pain, as an integrative and rehabilitative intervention of percutaneous lavage in calcific rotator cuff tendinopathy, compared to the percutaneous lavage alone. Moreover, the tendon calcification dimension at the baseline, and the percentage of drainage of the lesion, were associated with functional performance recovery and pain reduction after the UGN intervention.

In a recent Cochrane systematic review, several approaches were tested in order to assess the efficacy of electrotherapy for rotator-cuff-disease-related pain as an integrative or alternative method to oral and intrarticular pharmacological treatments [15], reporting a low quality of evidence and a short-term benefit of electrotherapy over placebo in people with calcific tendinitis [15]. In this study, the main hypothesis was to verify the utility of BEN in the management of rotator cuff pain during the hyperalgic phase of calcific tendinopathy, as an integrative method used after percutaneous lavage in calcific rotator cuff tendinopathy. One of the main significative moments in the formation of calcific lesions is the deposit in vesicles of calcium-hydroxyapatite crystals scattered between the chondrocytes and the fibro-cartilage tissue free of vascularization [4]. During this phase, the increased dimension of the lesion can elicit the nociceptor, determining a functional limitation of the shoulder with severe pain. Vascular neoformation characterized the third stage, where pain can also be spontaneously reduced. BEN was reported to ameliorate blood flow in this area. The increased vascularization was demonstrated in both the arterial and venous circle, and on skin microvascular perilesional perfusion tissue [21]. During BEN treatment, several changes occur including vascularization by angiogenesis and fibroblast proliferation. Locally, a more vascular endothelial growth factor was expressed, with increased production of fibroblast growth factor and an upregulation of anti-inflammatory genes [22].

LIFE-Stim™ sequences generated impulses administered transcutaneously; those stimuli are not periodic, thus avoiding the phenomenon of adaptation of biological tissues. They are able to implement synchronisms and rhythms in excitable structures by activating a mechanism of functional “restoration” of the area involved in the treatment. In conclusion this procedure is safe, and well tolerated by individuals, as demonstrated by the fact that no adverse events occurred during this study; however, no severe complications are to be expected if the procedure is performed as recommended.

In summary, to the best of our knowledge, this represents the first study using a percutaneous mini-invasive procedure, US-guided, lavage of the intratendinous calcification of the shoulder with the use of three needles, and with a positioning of a connector between those. This approach facilitates the achievement of intratendinous calcification and it reduces the possibility of the drainage obstruction, and most importantly, the connector ensures a more stable operating field for the operator. 

### Limitations and Strengths of the Study

Among limitations of the study, the small sample size needs to be considered. Moreover, the two treatment samples are not balanced in the SPADI score and self-perceived pain NRS score at the baseline which could have introduced a selection bias. In fact, an inadequate allocation concealment could exaggerate the estimation of treatment effect, on average [23]. Additionally, a performance bias could be introduced in this study, but the blind process for the operator is problematic to achieve despite the fact that we tried to balance this last bias, masking the group treatment during the statistical analytic approach. Among the many tools that could be used in the assessment of shoulder pain and related disability, we used the NRS score and the SPADI; those scales could be self-reported and are largely adopted in several studies, enabling the results comparison among different cohorts. Potentially, the use of three needles could expose patients to a higher risk of fragmentation of the calcification, but in our study and in our routinely clinical experience this event never happened.

Finally, a long-term follow-up could be useful in verifying the effectiveness of the combined treatment over time. One of the key strength points that has to be considered is the unicity of the study that, to the best of our knowledge, for the first time reported the effect on physical function and self-reported pain level of a lavage and drainage of tendon calcification of the shoulder, with a step-by-step standardization of the protocol, integrated by BEN.

## 5. Conclusions

BEN, as an integrative methodology of percutaneous lavage of calcific tendon of the shoulder, can increase physical function recovery and pain relief in a short period (15 days), with benefits lasting in the long period (40 days), compared to standard of care. Percutaneous lavage of calcific tendon is a simple approach which requires a trained operator to perform the procedure. In addition, a minimum of 40% drainage of the initial dimension of the calcific lesion is sufficient to produce a beneficial clinical effect.

## Figures and Tables

**Figure 1 ijerph-19-05837-f001:**
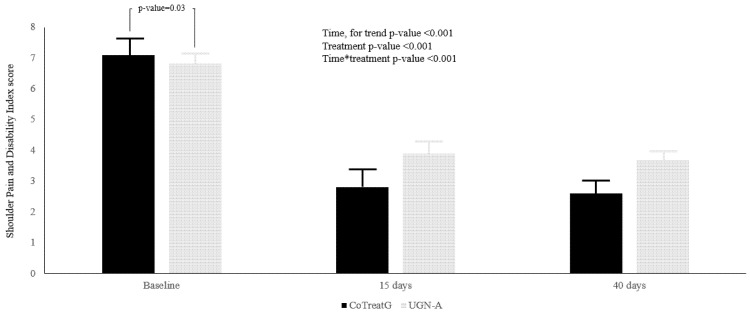
Shoulder Pain and Disability Index score was reported as mean ± S.E. The black bars represent the UGN plus BEN treatment group (CoTreatG), whereas gray bars represent the UGN-alone treatment (UGN-A). The differences between treatment were assessed with Linear Mixed Models, and were reported by the *p*-value for: Time, Treatment and the interaction between Time × Treatment.

**Figure 2 ijerph-19-05837-f002:**
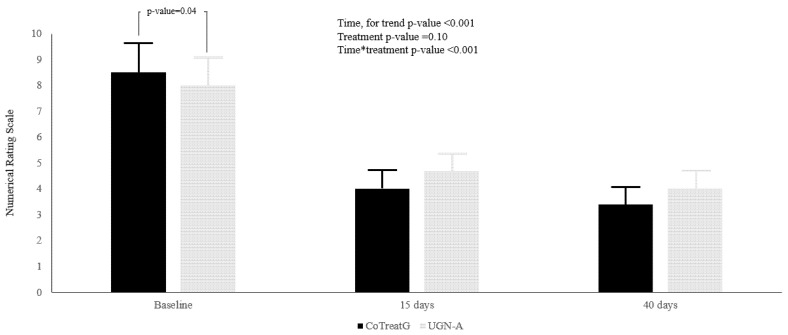
Visual Analogue Scale score was reported as mean ± S.E. In black histograms the UGN plus BEN treatment group (CoTreatG) is represented, whereas in gray histograms the UGN-alone treatment (UGN-A) is represented. The differences between treatments were assessed with Linear Mixed Models, and were reported by the *p*-value for: Time, Treatment and the interaction between Time × Treatment.

## Data Availability

The datasets used and/or analyzed during the current study are available from the corresponding author on reasonable request.

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
