# Peer review of "Efficacy of Ultrasound-Guided Percutaneous Lavage and Biocompatible Electrical Neurostimulation, in Calcific Rotator Cuff Tendinopathy and Shoulder Pain, A Prospective Pilot Study"

_ijerph, 2022, doi:10.3390/ijerph19105837_

Round 1

Reviewer 1 Report

"The Authors presented a very interesting study on the impact of the US-guided needling plus Biocompatible Electrical Neurostimulation in the treatment of the calcific tendinopathy of the shoulder. The material was homogenous. Forty adult patients (aged 40-60 years) with a diagnosis of CTS, in the acute, and colliquative phase, were recruited and enrolled into the study. The methodology was well prepared. Participants were assessed for self-perceived pain, and for functional limitation through the Shoulder Pain and Disability Index score. The authors presented reliable and significant results that Biocompatible Electrical Neurostimulation is a useful tool and increases functional performance and reduces shoulder pain in individuals with CTS. However, some minor language editing of the manuscript is required.

The specific comments please find below:

  • Line 28: instead of “40 adults” should be “Forty adults”
  • Line 31: please explain the abbreviation: “NRS”
  • Line 36: there is no need to repeat: “15 days after the enrollment”; I would suggest: “At the time T0”
  • Line 37: as above
  • Line 132: I would suggest using “posterior acoustic shadowing” instead of “poor shadow cone”
  • Line 164: The abbreviation “NPRS” is different from the abstract's “NRS”. Please use one abbreviation in the entire manuscript.
  • Line 172: Why the abbreviation UGN repeated?
  • Line 258: Why the abbreviation BEN repeated?
  • Line 277: Abbreviation VEGF is never used again and should be deleted."

Author Response

We thank the reviewer for the suggestions, and accordingly we have emended the text.

  • Line 28: instead of “40 adults” should be “Forty adults”
  • Line 31: please explain the abbreviation: “NRS”
  • Line 36: there is no need to repeat: “15 days after the enrollment”; I would suggest: “At the time T0”
  • Line 37: as above
  • Line 132: I would suggest using “posterior acoustic shadowing” instead of “poor shadow cone”
    • We thank this referee for the suggestion and we have emended the definition according to Sconfienza classification.
  • Line 164: The abbreviation “NPRS” is different from the abstract's “NRS”. Please use one abbreviation in the entire manuscript.
  • Line 172: Why the abbreviation UGN repeated?
  • Line 258: Why the abbreviation BEN repeated?
  • Line 277: Abbreviation VEGF is never used again and should be deleted."

Reviewer 2 Report

The manuscript “Efficacy of Ultrasound-guided percutaneous lavage and Biocompatible Electrical Neurostimulation, in calcific rotator cuff tendinopathy and shoulder pain, a prospective pilot study.” Authored by Pellegrino et al. represents an interesting clinical trial to deal with a calcification tendinopathy of the shoulder (CTS) that exhibits elevated incidence affecting patients’ lifestyle. The authors performed a comparative study to assess the implication of electrotherapy in improving shoulder recovery. The authors applied two treatment approaches consisting of UGN and a combination of UGN and BEN to deal with CTS. They found that the combinational effect between UGN and BEN improved patients’ functional recovery and reduced shoulder pain within the treated subjects. The work is well designed and well conducted. Before considering it for publications, the authors are invited to answer to some inquiries.

  • In line 51, the authors talked about “two forms of CTS” and they only talked about “the enthesopatic”. Please discuss about the second form.
  • Lines 230-232, the authors declare that 40 patients were included in the study with maintaining gender balances within groups. However, in the mentioned lines the authors stated “Also, genders were balanced between the two groups 11 (55.0%) and 13 (65.0%) for treatment and control respectively (p-value=0.56).” These numbers do not reflect the total patient numbers and the written sentence is confusing. Please discuss.

Author Response

We appreciate the suggestion of this Reviewer and accordingly we have emended the text:

In line 51, the authors talked about “two forms of CTS” and they only talked about “the enthesopatic”. Please discuss about the second form.

  • There are two forms of CTS: the enthesopathic, and the intratendinous. The enthesopathic form presents calcifications …

Lines 230-232, the authors declare that 40 patients were included in the study with maintaining gender balances within groups. However, in the mentioned lines the authors stated “Also, genders were balanced between the two groups 11 (55.0%) and 13 (65.0%) for treatment and control respectively (p-value=0.56).” These numbers do not reflect the total patient numbers and the written sentence is confusing. Please discuss.

  • No statistically significant differences could be found for sex between the two groups, females were 11 (55.0%) and 13 (65.0%) for treatment and control respectively (p-value=0.56).

Reviewer 3 Report

Pag 2 line 85: Please specify wich classification did you use.

Page 3 line 125: Please better specify the group of patients recruited in the study. You wrote “both in the acute and colliquative phase”. In my opinion it is better to use Martinoli or Sconfienza classifications.

Page 4 line 183: Albeit it is my opinion, I do not think that the use of 3 needles into the calcification is the best technique, especially in the case of soft calcification. Using 3 needles inside the calcification increases the possibility of fragmentation.

Line 188: Please specify the temperature of the saline solution used.

Page 7 line 263: You write that several approaches were tested. Please reported the result of these studies.

Author Response

Pag 2 line 85: Please specify which classification did you use.

We appreciate the suggestion and we have emended the definition of the lesion and accordingly we also insert a new citation.

Page 3 line 125: Please better specify the group of patients recruited in the study. You wrote “both in the acute and colliquative phase”. In my opinion it is better to use Martinoli or Sconfienza classifications.

We thank the reviewer for the suggestion, and accordingly we described the definition of Sconfienza, therefore the sentence now appear:

“Between March 2019 and November 2020, adult patients aged 40 to 60 years, with a diagnosis of CTS, in fluid calcifications, hypo or anechoic, without acoustic shadowing, were recruited and enrolled into the study (stage III according to Sconfienza classification) [10].”

Page 4 line 183: Albeit it is my opinion, I do not think that the use of 3 needles into the calcification is the best technique, especially in the case of soft calcification. Using 3 needles inside the calcification increases the possibility of fragmentation.

We thank the reviewer for the suggestion, but in our routinely clinical experience we did not have any fragmentation of the calcification, nor this event could be registered during the study. We have also search in the medline looking at any report of such event, but we found only one report from Giacomoni and Silotto with one needle. In this paper they are looking at a fragmentation as a therapeutical approach. Anyway, we considered you suggestion and in the limitation section we include the potential risk of fragmentation.

Potentially the use of three needles could expose patients to a higher risk of fragmentation of the calcification, but in our study and in our routinely clinical experience this event never happened.

Line 188: Please specify the temperature of the saline solution used.

We appreciate your suggestion, and accordingly we modify the text.

Page 7 line 263: You write that several approaches were tested. Please reported the result of these studies.

We appreciate your suggestion and accordingly we change the text that now appear as:

In a recent Cochrane systematic review, several approaches were tested in order to assess the efficacy of electrotherapy for rotator cuff disease related pain as an integrative or alternative method to oral and intrarticular pharmacological treatments [15], reporting a low quality evidence, and a short-term benefit of electrotherapy over placebo in people with calcific tendinitis [15].

Reviewer 4 Report

Reviewer comments:

Calcific tendinopathy of the shoulder (CTS) is the most common cause of shoulder pain. There is a general agreement that conservative treatment, such as US-guided needling (UGN) and Biocompatible Electrical Neurostimulation (BEN), is the first therapeutic choice for CTS. In this study, the main objective is to assess the effect of UGN alone and the combined treatment of UGN plus BEN, in the treatment of calcific rotator cuff tendinopathy and shoulder pain. The results indicate that UGN plus BEN improves functional performance and reduces shoulder pain in individuals with CTS.

There are still some points that the authors should consider, as described in the following. Also, some suggestions are provided, in case the authors consider them interesting to carry out.

The main objective of this study is to assess the effect of UGN (standard of care), compared to UGN plus BEN (treatment-group), in the treatment of calcific rotator cuff tendinopathy and shoulder pain. Therefore, the authors designed two groups: UGN plus BEN (combined-treatment-group, CoTreatG) and UGN-alone (UGN-A). Why not add another group of BEN-alone (BEN-A) in the comparison since BEN is also a possible conservative intervention in the hyperalgic phase?

In Materials and Methods line 151-153, please explain and provide more details. What are the curves of FAR and FRR to determine the CER? Is the definition of CER the same as EER? If yes, why not use the same term? If not, what is the difference between them? Please explain “CER of two points”, “EER of three points”, parameter α-error and parameter β-error.

Please note that some abbreviations are used without explanation or background about their meaning (e.g., CER, EER), which disrupts the reading flow. Such things rely on the reader's background, and though they may be understandable, a concise reference to what they mean is recommended, at least in an appendix towards the end. Please check the entire manuscript and properly use the abbreviations.

In 2.1 Outcome Measures, two different methods (SPADI and NPRS) are used to assess the pain and/or disability. There are still other scales to measure the shoulder function, e.g., the Disabilities of the Arm, Shoulder, and Hand (DASH) questionnaire, the American Shoulder and Elbow Surgeons (ASES) score, and the Simple Shoulder Test (SST) and so on. Why did the authors select these two methods rather than others for assessment?

In 2.1 Outcome Measures line 164, “Pain was assessed by the use of the Numerical Rating Scale (NPRS score) [18] that variates between 0 and 10 cm with 0 indicating no pain and 10 cm indicating the worst possible pain”. It seems that the NPRS score is not mentioned in the reference [18]. What does “cm” mean in this sentence? A unit? centimeter? Do the authors want to say that “varies between 0 and 10”? Here “variate” is a noun, not a verb.

In the supplementary materials, figures should be noted as “Figure S1” or “Supplementary Figure 1” to avoid the confusion with the main figures. Please keep the format consistent in the entire manuscript, either “Figure S1” or “Supplementary Figure 1”, in both main text and supplementary.

In Results line 230-231, “Also, genders were balanced between the two groups 11 (55.0%) and 13 (65.0%) for treatment and control respectively (p-value=0.56).” Does it mean 11 and 13 women for treatment and control group, respectively?

In Figure 1 and Figure 2, the differences between treatment were assessed with Linear Mixed Models and were reported by the p-value for: time, treatment and the interaction between time*treatment. In addition to the p-value, the author should also report modeling analyses, e.g., intraclass correlation coefficient (ICC) or variance partitioning coefficient (VPC), or something else, to indicate the reliability of an experimental method.

In Figure 2, the treatment p-value is 0.1 in the report of Linear Mixed Models, indicating no statistically significant difference (p-value > 0.05). This doesn’t match the corresponding treatment p-value < 0.001 as shown in Figure 1. Please explain why there is no significant difference in the treatment in Figure 2 and what causes the mismatch between Figure 1 and Figure 2.

In Results line 242-245, “At the baseline the calcification sizes were very similar (CoTreatG: 1.43 ± 0.23 cm; BEN-A 1.38 ± 0.25 cm; p-value= 0.50); moreover, both at T1 and T2 evaluations, a statistically significant dimensions reduction, for all the patients enrolled in the study (-0.77±0.03; p-value<0.001) was found.” What is BEN-A here? When talking about the dimension reduction (-0.77±0.03; p-value<0.001), it that a reduction from T0 to T1 or from T0 to T2?

In Results line 245-246, “No multiplicative effect for the interaction between T0, T1 and T2 and treatment group were detected (β±SE;0.03±0.07; p-value for the interaction=0.70).” Dose β±SE represent mean and standard error? Is there any special reason to add the term “β±SE” here before the statistical result (0.03±0.07), but not in other statistical results?

In Results line 242-249, when reporting the calcification size reduction, it would be helpful to compare and analyze the results in some bar graphs, like Figure 1 and Figure 2.

Minor comments:

In Abstract line 23, there should be a period in the end of the second sentence “Conservative treatment is considered as the first therapeutic choice for CTS”.

In Abstract line 26-27, “Pilot, prospective, non-interventional, monocentric, observational study, of patients treated for calcific rotator cuff tendinopathy and shoulder pain.” This sentence doesn’t have a verb.

In Abstract line 35, “Thorough the study, both groups improved their perceived functional performance of the arm (p-value <0.001).” Does it mean “through the study” or “throughout the study”?

In Introduction line 51-53, “There are two forms of CTS: the enthesopathic, which presents calcifications typically localized at the tendon insertion, often associated with erosions of the tendons [3].” This sentence seems strange and incomplete. What are those two forms of CTS?

In Introduction line 58, “… seem to be gradually gaining acceptance”. “seem” should be “seems”.

In 2.1 Outcome Measures line 155, “At the enrolment day all subjects underwent a clinical and anamnestic evaluation, …”. Here “enrolment” is typo.

In the figure legend of Figure 1 and Figure 2, “In black histograms was represented the UGN plus BEN treatment group (CoTreatG) whereas, in gray histograms was represented the UGN alone treatment 194 (UGN-A).” A proper expression should be “the black bars represent …”, or “the UGN plus BEN treatment group (CoTreatG) is represented by the black bars”.

In the figure legend of Figure 1 and Figure 2, “The differences between treatment were assessed with Linear Mixed Models and were reported the p-value for: Time, Treatment and the interaction between time*treatment.” It should be “reported by the p-value”. If “Time” and “Treatment” are capitalized, “time*treatment” should be “Time*Treatment”.

In Results line 250-252, “The tendon calcification dimension was associated with both perceived functional performance of the arm by mean of SPADI-score (1.99±0.11; p-value<0.001) as the self-reported pain NRS-score (4.68 ± 0.30; p-value< 0.001) independently from group treatment and evaluation time”.  We should say “both … and …”, not “both … as …”.

Author Response

The main objective of this study is to assess the effect of UGN (standard of care), compared to UGN plus BEN (treatment-group), in the treatment of calcific rotator cuff tendinopathy and shoulder pain. Therefore, the authors designed two groups: UGN plus BEN (combined-treatment-group, CoTreatG) and UGN-alone (UGN-A). Why not add another group of BEN-alone (BEN-A) in the comparison since BEN is also a possible conservative intervention in the hyperalgic phase?

We appreciate the Referee suggestion. All patients were in a hyperalgic phase, and we are not interested in the property of BEN as a conservative instrument, but we want to know if BEN could be used as an integrative method after UGN. In any case in the text line 275-277 we clearly stated that we want to analyze the “plus-effect” of BEN in the CTS treatment. This could be an interesting suggestion for a new study that compare different conservative treatments.

In Materials and Methods line 151-153, please explain and provide more details. What are the curves of FAR and FRR to determine the CER? Is the definition of CER the same as EER? If yes, why not use the same term? If not, what is the difference between them? Please explain “CER of two points”, “EER of three points”, parameter α-error and parameter β-error.

We thank the reviewer for the suggestion, this was a typo and we have emended it, moreover according to the suggestion we change a little the text that now appears as: “0.05 α-error (Type-I), 0.20 β-error (Type-II).”

Please note that some abbreviations are used without explanation or background about their meaning (e.g., CER, EER), which disrupts the reading flow. Such things rely on the reader's background, and though they may be understandable, a concise reference to what they mean is recommended, at least in an appendix towards the end. Please check the entire manuscript and properly use the abbreviations.

We found very interesting this suggestion, therefore in the supplementary materials we insert a short table with a very concise definition of the acronymous used in the text.

In 2.1 Outcome Measures, two different methods (SPADI and NRS) are used to assess the pain and/or disability. There are still other scales to measure the shoulder function, e.g., the Disabilities of the Arm, Shoulder, and Hand (DASH) questionnaire, the American Shoulder and Elbow Surgeons (ASES) score, and the Simple Shoulder Test (SST) and so on. Why did the authors select these two methods rather than others for assessment?

We appreciate this suggestion, obviously SPADI and NRS are more familiar, for us. Moreover, could be used in a self-reported way, shortening the time of the assessment, and lastly, looking at the literature several studies use those scales. In accordance with this reviewer observation, we change the text that now appear as:

Line 310-313

“Among the many tools that could be used in the assessment of shoulder pain and related disability, we use the NRS-score, and the SPADI; those scales could be self-reported and are largely adopted in several studies, enabling the results comparison among different studies.”

In 2.1 Outcome Measures line 164, “Pain was assessed by the use of the Numerical Rating Scale (NRS score) [18] that variates between 0 and 10 cm with 0 indicating no pain and 10 cm indicating the worst possible pain”. It seems that the NPRS score is not mentioned in the reference [18]. What does “cm” mean in this sentence? A unit? centimeter? Do the authors want to say that “varies between 0 and 10”? Here “variate” is a noun, not a verb.

We thank the reviewer for the suggestion. The referenced paper, deals with almost all the scales used in the measurement in the shoulder pain and function. Moreover, we emended the text, that now appears:

Pain was assessed by the use of the Numerical Rating Scale (NRS score) [18] that vary between 0 and 10 with 0 indicating no pain and 10 indicating the worst possible pain.

In the supplementary materials, figures should be noted as “Figure S1” or “Supplementary Figure 1” to avoid the confusion with the main figures. Please keep the format consistent in the entire manuscript, either “Figure S1” or “Supplementary Figure 1”, in both main text and supplementary.

We appreciate the suggestion. We emended the text and the figure legend.

In Results line 230-231, “Also, genders were balanced between the two groups 11 (55.0%) and 13 (65.0%) for treatment and control respectively (p-value=0.56).” Does it mean 11 and 13 women for treatment and control group, respectively?

We thank the referee, and we apologies for the mistakes. We emended the text that now appears:

No statistically significant differences could be found for sex between the two groups, females were 11 (55.0%) and 13 (65.0%) for treatment and control respectively (p-value=0.56).

In Figure 1 and Figure 2, the differences between treatment were assessed with Linear Mixed Models and were reported by the p-value for: time, treatment and the interaction between time*treatment. In addition to the p-value, the author should also report modeling analyses, e.g., intraclass correlation coefficient (ICC) or variance partitioning coefficient (VPC), or something else, to indicate the reliability of an experimental method.

We appreciate the suggestion. We did not include those coefficients in previous version of the paper, we have now changed the text that now appear:

Moreover, seven percent in the NRS-score is attributable to differences between group (ρ=7.1%), whereas 41% of the within-person variation in NRS-score is associated with linear time (R2e=40.7).  

We also explained the meaning of those parameters in the statistical section.

In Figure 2, the treatment p-value is 0.1 in the report of Linear Mixed Models, indicating no statistically significant difference (p-value > 0.05). This doesn’t match the corresponding treatment p-value < 0.001 as shown in Figure 1. Please explain why there is no significant difference in the treatment in Figure 2 and what causes the mismatch between Figure 1 and Figure 2.

We thank the reviewer for his/her observation, but we did not agree. As matter of fact with a second or higher order equation, the most important element to see is the “higher order interaction term”; the first order terms could be also not statistically significant. Therefore, this analysis highlighted that the slopes of the two treatments are different, and this is evident in the p-value for the interaction, not in the p-value for the treatment.

In Results line 242-245, “At the baseline the calcification sizes were very similar (CoTreatG: 1.43 ± 0.23 cm; BEN-A 1.38 ± 0.25 cm; p-value= 0.50); moreover, both at T1 and T2 evaluations, a statistically significant dimensions reduction, for all the patients enrolled in the study (-0.77±0.03; p-value<0.001) was found.” What is BEN-A here? When talking about the dimension reduction (-0.77±0.03; p-value<0.001), it that a reduction from T0 to T1 or from T0 to T2?

We thank the reviewer for the observation. We reanalyze data and the change we reported referred to change in dimension from baseline (T0) to second follow-up (T2). Therefore accordingly we change the text

At the baseline the calcification sizes were very similar (CoTreatG: 1.43 ± 0.23 cm; BEN 1.38 ± 0.25 cm; p-value= 0.50); moreover, from T0 to T2 assessment, a statistically significant reduction in the dimensions, for all the patients enrolled in the study (-0.77±0.03; p-value<0.001) was found.

In Results line 245-246, “No multiplicative effect for the interaction between T0, T1 and T2 and treatment group were detected (β±SE;0.03±0.07; p-value for the interaction=0.70).” Dose β±SE represent mean and standard error? Is there any special reason to add the term “β±SE” here before the statistical result (0.03±0.07), but not in other statistical results?

We thank the reviewer, was a typo of the precedent version, we erase “β±SE” from the text.

In Results line 242-249, when reporting the calcification size reduction, it would be helpful to compare and analyze the results in some bar graphs, like Figure 1 and Figure 2.

We thank the reviewer for the suggestion; accordingly, we have inserted a new figure in supplementary material (Supplementary Figure 4).

Supplementary Figure 4: Variation across times of the study in the absolute dimension (cm) of the tendons calcification in the subjects enrolled. Statistical differences among times of the study were assessed using Linear Mixed Models.

Minor comments:

In Abstract line 23, there should be a period in the end of the second sentence “Conservative treatment is considered as the first therapeutic choice for CTS”.

We thank the reviewer for the suggestion; we have mended the text.

In Abstract line 26-27, “Pilot, prospective, non-interventional, monocentric, observational study, of patients treated for calcific rotator cuff tendinopathy and shoulder pain.” This sentence doesn’t have a verb.

We thank the reviewer for the observation, this is only a description, a definition of the study.

In Abstract line 35, “Thorough the study, both groups improved their perceived functional performance of the arm (p-value <0.001).” Does it mean “through the study” or “throughout the study”?

In Introduction line 51-53, “There are two forms of CTS: the enthesopathic, which presents calcifications typically localized at the tendon insertion, often associated with erosions of the tendons [3].” This sentence seems strange and incomplete. What are those two forms of CTS?

We thank the reviewer for the suggestion; we have emended the text.

There are two forms of CTS: the enthesopathic, and the intratendinous. The enthesopathic form presents calcifications

In Introduction line 58, “… seem to be gradually gaining acceptance”. “seem” should be “seems”.

We thank the reviewer for the suggestion; we have emended the text.

In 2.1 Outcome Measures line 155, “At the enrolment day all subjects underwent a clinical and anamnestic evaluation, …”. Here “enrolment” is typo.

We appreciate your suggestion and we erase “At the enrolment” and the sentence start with: “All subjects underwent …”

In the figure legend of Figure 1 and Figure 2, “In black histograms was represented the UGN plus BEN treatment group (CoTreatG) whereas, in gray histograms was represented the UGN alone treatment 194 (UGN-A).” A proper expression should be “the black bars represent …”, or “the UGN plus BEN treatment group (CoTreatG) is represented by the black bars”.

In the figure legend of Figure 1 and Figure 2, “The differences between treatment were assessed with Linear Mixed Models and were reported the p-value for: Time, Treatment and the interaction between time*treatment.” It should be “reported by the p-value”. If “Time” and “Treatment” are capitalized, “time*treatment” should be “Time*Treatment”.

In Results line 250-252, “The tendon calcification dimension was associated with both perceived functional performance of the arm by mean of SPADI-score (1.99±0.11; p-value<0.001) as the self-reported pain NRS-score (4.68 ± 0.30; p-value< 0.001) independently from group treatment and evaluation time”.  We should say “both … and …”, not “both … as …”.

We thank the reviewer for those suggestions; we have emended the text accordingly, and highlighted the corrections.